# Acquisition and maintenance of disgust reactions in an OCD analogue sample: Efficiency of extinction strategies through a counter-conditioning procedure

**Caroline Novara**[1,2☯]*, **Cindy Lebrun**[1☯], **Alexandra Macgregor**[3‡], **Bruno Vivet**[2], **Pierre Thérouanne**[4‡], **Delphine Capdevielle**[3], **Stephane Raffard**[1,3☯]

1 Univ Paul Valéry Montpellier 3, Univ. Montpellier, EPSYLON EA 4556, Montpellier, France, 2 Groupe Ramsay Gds, Clinique RECH, Montpellier, France, 3 Service Universitaire de Psychiatrie Adulte, CHU Montpellier, Montpellier, France, 4 Université Côte d'Azur, LAPCOS (EA 7278), Nice, France

☯ These authors contributed equally to this work.
‡ These authors also contributed equally to this work.
* caroline.novara@icloud.com

**Data Availability Statement:** All relevant data are within the paper and its Supporting Information files.

## Abstract

### Background

Obsessive-compulsive disorder (OCD) has long been considered as an anxiety disorder, disgust is the dominant emotion in contamination-based OCD. However, disgust seems resistant to exposure with response prevention partly due to the fact that disgust is acquired through evaluative conditioning.

### Aims

The present research investigates a counter-conditioning intervention in treating disgust-related emotional responses in two groups of individuals with high (High contamination concerns, HCC, n = 24) and low (Low contamination concerns LCC, n = 23) contamination concerns.

### Methods

The two groups completed a differential associative learning task in which neutral images were followed by disgusting images (conditioned stimulus; CS+), or not (CS-). Following this acquisition phase, there was a counter-conditioning procedure in which CS+ was followed by a very pleasant unconditional stimulus while CS- remained unreinforced.

### Results

Following counter-conditioning, both groups reported significant reduction in their expectancy of US occurrence and reported less disgust with CS+. For both expectancy and disgust, reduction was lower in the HCC group than in the LCC group. Disgust sensitivity was highly correlated with both acquisition and maintenance of the response acquired, while US expectation was predicted by anxiety.

**Funding:** CN CIFRE device from French ministry of research and industry.

**Competing interests:** The authors have declared that no competing interests exist.

**Abbreviations:** OCD, Obsessive-compulsive disorder; C-OCD, Contamination related obsessive compulsive disorder; EC, Evaluative conditioning; EPR, exposition with response prevention; DS, Disgust sensitivity; DP, Disgust propensity; US, unconditional stimulus; CS, conditional stimulus; HCC, High dirtiness and contamination concerns; LCC, Low dirtiness and contamination concerns; CC, counter-conditioning.

## Conclusion

Counter-conditioning procedure reduces both expectations and conditioned disgust.

## Introduction

OCD is a psychological disorder whose prevalence is estimated between 1.5% and 3.5% of the world population. This disorder affects all cultural and ethnic groups and is considered by the World health organization as one of the 10 most disabling mental health disorders [1]. Behavioral and pharmacological treatments for OCD have been repeatedly demonstrated to be effective, leading to large and sustained reductions in OCD symptoms [2]. The Cognitive Behavioral Therapy program with the most empirical support for its efficacy is exposure with response prevention (ERP), even in the most severe cases [3]. However, even if ERP is an evidence-based treatment for treating OCD, not all patients achieve treatment successfully and many patients continue to experience moderate to high levels of symptoms and/or co-occurring behavioral and emotional problems following treatment (75 to 80% of patients respond, only 40 to 52% achieve remission) [4].

From a theoretical point of view, ERP is based on the cognitive-behavioral model in which the mechanisms of fear acquisition through Pavlovian Conditioning play an important conceptual role in explaining the development and the maintenance of the symptoms [5]. It has been proposed that the main reasons of ERP's failure include patients' lack of motivation to reduce rituals and the presence of comorbid disorders, such as depression or avoidant personality, specifically in contamination-based OCD [6].

In addition to these variables, recent research has pointed out the key role of various negative emotions such as shame, embarrassment, frustration, anger and contempt and more particularly disgust in the maintenance of contamination-related OCD [7]. Nonetheless, until recently, how disgust is acquired and how it can be extinguished has received very little empirical attention from researchers working in the development of more effective strategies for treating OCD [8].

Disgust is a response to stimuli that have the ability to soil, and are considered threatening, polluted or of little value [9]. Disgust has long been under-researched compared to other emotions [10], such as anger, fear or sadness. Since the '90s however, authors have examined this emotion further and it is clear today that it is involved in a wide range of disorders previously conceptualized within the exclusive framework of anxiety. Examples of such disorders are specific phobias, especially fear of spiders [11], phobias of blood and injections [12], post-traumatic stress disorder [13] and more particularly contamination related OCD (C-OCD) [14,15]. Current research indicates that behind the word « disgust » two notions must be distinguished: Disgust Sensitivity (DS, i.e how aversive it is for an individual to feel disgust) and Disgust Propensity (DP, i.e the tendency to experience disgust intensely and frequently) [16,17]. Regarding, the relationship between disgust and OCD contamination symptoms, high disgust propensity was significantly increased in individuals with C-OCD compared to individuals with other subtypes of OCD [18]. Studies have also reported that increased sensitivity to disgust was strongly correlated with self-reported contamination symptoms [19].

Despite increased evidence of the contribution of disgust to C-OCD [20], the literature remains unclear on the processes underlying disgust extinction. Indeed, there are compelling arguments taken from laboratory studies showing that, in contrast to fear, disgust is resistant to extinction and that exposure treatments are less effective in reducing disgust than fear [21,22]. If it is now well established that anxiety is acquired through conventional Pavlovian

conditioning mechanisms, other emotions do not exclusively respond to this type of learning. Despite the small number of studies that have studied disgust in the conceptual framework of conditioning theories [21–24] qualitative differences with other forms of Pavlovian conditioning have been identified, including the reference to a "hedonic shift" from the unconditional stimulus (US) to the conditional stimulus (CS) [25]. A learning model has been proposed in which abnormal disgust reactions can be maintained through the process of evaluative conditioning (EC) [8]. EC involves the transfer of valence from an unconditioned stimulus to a conditioned stimulus so that the conditioned stimulus acquires the aversive characteristics of the unconditioned stimulus regardless of its relation to the unconditioned stimulus.

Even if there is considerable evidence to support, similarly to fear, the role of traditional classical predictive conditioning mechanisms in the acquisition of disgust, recent research has suggested that evaluative conditioning effect (EC) could be central in the learning of disgust [21–28]. Although EC procedure looks like Pavlovian conditioning, it differs in being less susceptible to extinction [22,26–28]. Whereas prediction is central in Pavlovian conditioning, it's not the case in EC which is characterized by transfer of affective valence. It has in fact been argued that reactions of disgust are essentially acquired by evaluative conditioning effects: an associative learning process by which a neutral stimulus takes on an affective label following its association with an unconditional stimulus, positively or negatively valued [29–31]. Evaluative conditioning, by allowing individuals to evaluate both new and familiar emotional stimuli (eg, whether something is liked or not, disgusting or desirable) thus facilitates the role of disgust not only as a survival mechanism [24] but also as a fundamental adaptation mechanism of humans to their environment.

If disgust is relatively resistant to extinction due to evaluative conditioning, it may be useful to include disgust-focused exposure exercises in treatment [32] or additional exposure trials [33]. In the treatment of blood-injection-injury phobia, targeted disgust-based exposure achieves better symptom reduction compared to traditional exposure based on anxiety [34]. Research also suggests that approaches such as reappraisal of unconditional stimulus or counter-conditioning may be useful alternatives to traditional exposure when targeting reduction of affective conditioned responses to reduce both valence and expectation issues [29]. However, research has still to explore whether conditioned disgust responds to these approaches.

In this study, we decided to focus on the potential utility of a counter-conditioning procedure to reduce learned disgust reactions. The counter-conditioning procedure implies pairing a conditioned stimulus (CS) with an unconditional stimulus (US) of an opposite valence from the original US. This technique could be effective for two reasons. First, by reducing the effects of the negative evaluation attributed to an item by granting it the valence of the new representation. Secondly, by reducing the expectation of the conditioned response as it can be through simple exposure techniques by replacing the old aversive association by a new one. Several studies using self-reported measures, emotional priming tasks, as well as behavioral tasks have shown positive effects of counter-conditioning on disgust-related evaluative learning [35,36]. However, despite the fact that the participants underwent a conditioning procedure with disgusting unconditional stimuli, these authors did not directly assess the disgusting aspect of their conditioned stimuli, limiting the scope of the conclusions that can be drawn from it.

Our main objective was to test the efficiency of a counter-conditioning procedure on a response of acquired disgust in two groups of participants high (HCC) and low (LCC) in contamination concerns, using a differential associative learning task that dissociated predictive learning from evaluative learning. The second objective was to observe if disgust sensitivity and propensity predict the valence attributed to the conditioned stimulus. We hypothesized that the group with high concerns will (a) score significantly higher on the disgust scale and

(b) be more prone to acquired disgust response and less able to disengage from the evaluative learning. Furthermore we assumed (c) that the counter-conditioning procedure would succeed in reducing US expectation and the valence attributed to the CS.

## Materials and methods

### Participants

Following Amstrong et al.'s procedure (2017), one hundred and sixty-two undergraduate students from Paul Valery Montpellier 3 University were tested using Padua's inventory contamination subscale (PI-C) [37] to identify people with low and high contamination-related concerns. In line with other studies [38–40], individuals with scores greater than 13 (77: 47.53% of the sample tested) were considered eligible for the group with high concerns about dirt and contamination (HCC) and individuals with scores below 7 (42: 27.92% of the sample tested) were considered eligible for the group with low concerns regarding dirt and contamination (LCC). In order to maximize the distinction between groups we then started recruiting participants with highest and lowest scores. Given the timeline of our study, our final sample comprised 24 participants in the HCC group and 23 participants in the LCC group.

### Measures

In addition to the PI-C measure used to allocate participants to experimental groups, we used various measures to assess disgust tendencies (disgust propension and sensibility scale (DPSSRf-10), [41]), obsessive-compulsive symptomatology (dimensional obsession compulsion scale (DOCS) [5], obsession compulsion inventory (OCI-R) [42]), anxiety (state and trait anxiety inventory (STAI B) [43]), and positive and negative affect schedule (PANAS) [44].

We also used the unipolar version of the empirical valence scale (EVS) [45] to collect perceived disgust and US expectation during the task. EVS is a labeling scale designed to rate the magnitude of subjective experiences. Participants rate their disgust in response to the CS and US, and their US expectation during the CS presentation. The unipolar version of the scale contains the following labels and associated values: not at all (0), barely (7), slightly (12), averagely (24), moderately (38), strongly (70), extremely (85), the most imaginable (100). These labels are placed on a line (without the corresponding numeric values). Scores are made by clicking on the line using a mouse.

### Material

The CSs consisted of two neutral abstract images, respectively pink and blue, representing lines and circles (480x480p.). CS allocation was counter-balanced between participants in each group so that in one condition, for half of the participants of the whole group (N = 24), image #1 was CS +, and image #2 was CS- and for the other half vice versa. The US consisted of eight different disgusting images (824x618p.) depicting spoiled food (2), waste/body secretions (3), unhygienic environments (3) that have been selected from the DIRTI database [46], and data resources online. The same eight US were used for each participant. We used multiple US to limit habituation. The stimuli were presented on a screen using the Eprime 2.0 software control stimuli presentation and data collection.

### Procedure

The Ethics Committee of the Psychological Faculty of Montpellier 3 University approved the study, and all participants provided written informed consent. Participants then

completed the measures (PI-C, DPSS-R, DOCS) as well as a collection of general informa-
tion (age, gender, level study, professional status). The participants then completed the
computer task, adapted from the work of Armstrong & Olatunji (2016) which consisted of
three phases:

**Habituation phase.**    First, participants see eight non-reinforced presentations (6 s) from
each CS. CS are preceded by a cross (1.5 s) and followed by an inter trail interval (ITI) that var-
ies between 12 and 18 seconds. CS appears in the center of the screen.

**Acquisition phase.**    CSs are presented for 6 seconds in the center of the screen. Immedi-
ately after the presentation of the CS +, the US is presented for 3 sec. After the presentation of
the CS-, the test continues with the ITI. CSs are preceded by a fixation cross (1.5) and followed
by an ITI that varies between 12 and 18 sec. There are eight presentations of the CS- and 8 pre-
sentations of the CS +, in pseudo-random order (no more than four consecutive presentations
of the same CS).

**Counter conditioning phase.**    The acquisition procedure is repeated but CS+ is followed
by very pleasant US extracted from the IAPS and public online resources, while CS- remains
unreinforced. Pleasant US were eight images representing landscapes (2), family (1), animals
(2), a baby (1), laughing children (2).

At the end of each phase, participants use the empirical valence scale (EVS) to score the dis-
gust felt for CS "How much do you find this picture disgusting?" and how much they expect
the disgusting US to follow CS "How much do you expect this picture to be followed by a dis-
gusting picture". For each rating, stimuli and scale are presented simultaneously until a
response is given.

## Statistical plan

All statistical analyses were performed with IBM SPSS software version 24.0. Following the
work lead by Armstrong and Olatunji in 2016, analyses were conducted separately for each
phase of the procedure (habituation, acquisition, counter conditioning (CC)). Consequently,
we did not correct for multiple comparison across phases. For the disgust ratings and the US
expectancy ratings, we used mixed-model repeated analysis of variance with Group (HCC;
LCC) as between-subject factor and CS (CS+; CS-) as the within-subject factor. Significant CS
x group interaction were followed by independent sample t-test in order to compare between
groups the ratings of the CS+ and the CS- separately. To observe the evolution of the ratings
between acquisition and counter conditioning stages, a same statistical design was applied add-
ing the within-subject factor of phase (acquisition, counter conditioning) to the previous
ANOVA model that included the CS and group factors. Phase by CS interaction was examined
to observe the decrease in disgust and US expectancy ratings. Using paired sample t-test we
probed this interaction by contrasting the CS+ and the CS- between acquisition and counter
conditioning. The phase by CS by group interaction was also examined. We conducted inde-
pendent sample t-test comparing the groups on changes scores in which responding at CC was
subtracted from responding at acquisition for each CS. Then two regression analyses were per-
formed to measure the predictive value of Disgust propensity and sensitivity, contamination
concerns (OCI-R contamination, DOCS Contamination, and PADUA Contamination), nega-
tive affects (PANAS negative) and anxiety trait scores (STAI B) on US expectancy ratings of
the conditioned CS and disgust ratings of the acquisition phase. Since we have seven predictors
in each of our two regression analyses, we chose to use a Bonferroni correction to correct for
multiple comparisons. For this, we divided our $\alpha$ value (.05) by 7 to get a more realistic signifi-
cance level (i.e. .007).

All analyses were conducted with a significance threshold of $\alpha \leq .05$, two-tailed.

## Results

### Group characteristics

Groups with high contamination concerns (HCC) and low contamination concerns (LCC) did not show significant differences in age, gender, and socio-professional category. As expected, the HCC group had significantly higher levels of behavioral contamination-related symptoms (PI-C: p < .001; DOCS contamination p < .001; OCI-R contamination p < .001), was higher in sensitivity and disgust propensity (p < .001) and had significantly higher levels of anxiety (p = .004) and negative affect (p = .040). The HCC group also rated the disgusting US after the acquisition phase as significantly more disgusting than the LCC group did (p < .001) (See Table 1).

### Habituation phase

**Unconditional stimulus expectation rating.** The main effects of CS, $F(1,43) = 2.15$, $p = .15$, $n^2_p = .04$, the main effect of group, $F(1,43) = .02$, $p = .87$, $n^2_p = .001$, and the interaction between the group and CS $F(1,43) = 0.43$, $p = .51$, $n^2_p = .01$ did not reach significance. Thus, there were no differences before acquisition in the susceptibility of CSs to provoke US expectation for none of the groups (Table 2, Fig 1).

**Disgust ratings.** The main effects of CS $F(1,43) = 1.04$, $p = .31$, $n^2_p = .02$, group $F(1,43) = .03$, $p = .17$, $n^2_p = .04$, and group interaction with the CS $F(1,43) = .42$ $p = .61$, $n^2_p = .006$, were all insignificant. Thus, before the acquisition, CSs do not differ in their ability to elicit disgust (Table 2, Fig 1).

### Acquisition phase

**US expectation ratings.** The main effect of CS was significant $F(1,43) = 2206.36$, $p < .001$; $n^2_p = .98$. Participants rated the US expectancy occurrence significantly higher for CS + than for CS-, so participants learned contingency during the acquisition process. The main effect of group $F(1,43) = .50$, $p = .48$, $n^2_p = .01$, and the interaction between the CS and the group $F(1,43) = 1.27$, $p = .26$, $n^2_p = .02$ were both insignificant, indicating that there was no difference

**Table 1. Group characteristics.**

|  | HCC (N = 24) | LCC (N = 23) | 95% CI | Test Statistic, Sig. |
|---|---|---|---|---|
| **Age (years)** | 31 (11.86) | 37 (14.96) | [-13; 99] | $\chi^2 = -1.4$, p = .15 |
| **Genre (% Female)** | 68.18 | 60.86 |  | $\chi^2 = .26$, p = .60 |
| **0ccupation (% Students)** | 40.90 | 30.43 |  | $\chi^2 = 4.36$, p = .35 |
| **MEASURES** |  |  |  |  |
| **PI-C** | 22.43 (4.78) | 3.89 (1.99) | [16.09; 20.99] | t(43) = 15.39, p < .001 |
| **DPSSf-10 Propension** | 20.15 (2.87) | 15.43 (3.66) | [3.03; 7] | t(43) = 5.09, p < .001 |
| **DPSSf-10 sensibility** | 10.31 (3.95) | 6.60 (2.27) | [1.78; 5.63] | t(43) = 3.87, p < .001 |
| **DOCS Contamination** | 7.81 (3.06) | 3.30 (1.76) | [3.01; 6.01] | t(43) = 6.08, p < .001 |
| **OCI-R Contamination** | 5(2.07) | .35(.71) | [3.72; 5.57] | t(43) = 10.16, p < .001 |
| **STAI-B** | 49.33(10.10) | 40.67(8.38) | [2.87; 14.45] | t(43) = 3.02, p = .004 |
| **PANAS negative affect** | 15.92(3.27) | 14.04 (2.68) | [.08; 3.68] | t(43) = 2.11; p = .04 |
| **Mean US evaluation** | 76.96 (9.57) | 49.87 (2.97) | [16.69; 37.47] | t(43) = 5.25, p < .001 |

Note: Mean (SD); HCC: High contamination concerns group, LCC: Low contamination concerns group. DPSSf-10: disgust propension and sensibility scale, DOCS: Dimensional obsessional compulsion scale; OCI-R: Obsession compulsion inventory revised; STAI-B: State and trait anxiety inventory: PANAS: Positive and negative affect schedule).

**Table 2. Mean (SD), and between-group comparisons of direct evaluation on US expectation and valence ratings on empirical valence scale (100 points).**

|  |  | HCC (N = 24) | LCC (N = 23) | Test Statistic | Sig., effect size |
|---|---|---|---|---|---|
| **EXPECTATION** |  |  |  |  |  |
| **Habituation** | CS+ | 31.13 (32.10) | 35.69 (30.79) | $F (1,43) = .02$ | $p = .87, n^2_p = .001$ |
|  | CS- | 41.81 (33.98) | 39.78 (28.61) |  |  |
| **Acquisition** | CS+ | 91.13 (11.01) | 94.69 (14.37) | $F (1,43) = .50$ | $p = .48, n^2_p = .01$ |
|  | CS- | 2.09 (2.36) | 1.26 (1.25) |  |  |
| **Counter conditioning** | CS+ | 31.59 (32.42) | 11.39 (20.86) | $F (1,43) = 2.32$ | $p = .13, n^2_p = .07$ |
|  | CS- | 7.31 (15.56) | 10.34 (20.85) |  |  |
| **DISGUST** |  |  |  |  |  |
| **Habituation** | CS+ | 5.18 (14.75) | 1.26 (1.25) | $F (1,43) = .03$ | $p = .17, n^2_p = .04$ |
|  | CS- | 5.95 (14.75) | 1.52 (1.72) |  |  |
| **Acquisition** | CS+ | 11.18 (18.09) | 2.78 (5.06) | $F (1,43) = 4.75$ | $p = .03, n^2_p = .10$ |
|  | CS- | 1.81 (2.10) | 1.26 (1.25) |  |  |
| **Counter conditioning** | CS+ | 8.22 (16.46) | 1 (0) | $F (1,43) = 12.98$ | $p = .001, n^2_p = .23$ |
|  | CS- | 1.81 (2.10) | 1 (0) |  |  |

HCC: High contamination concerns group, LCC: Low contamination concerns group. CS-: Unconditioned stimulus; CS+: Conditioned stimulus.

in groups in potential predictability of unconditional stimulus occurrence following CS + (Table 2, Fig 1). Both groups learned the predictive association between the CS + the and disgusting pictures.

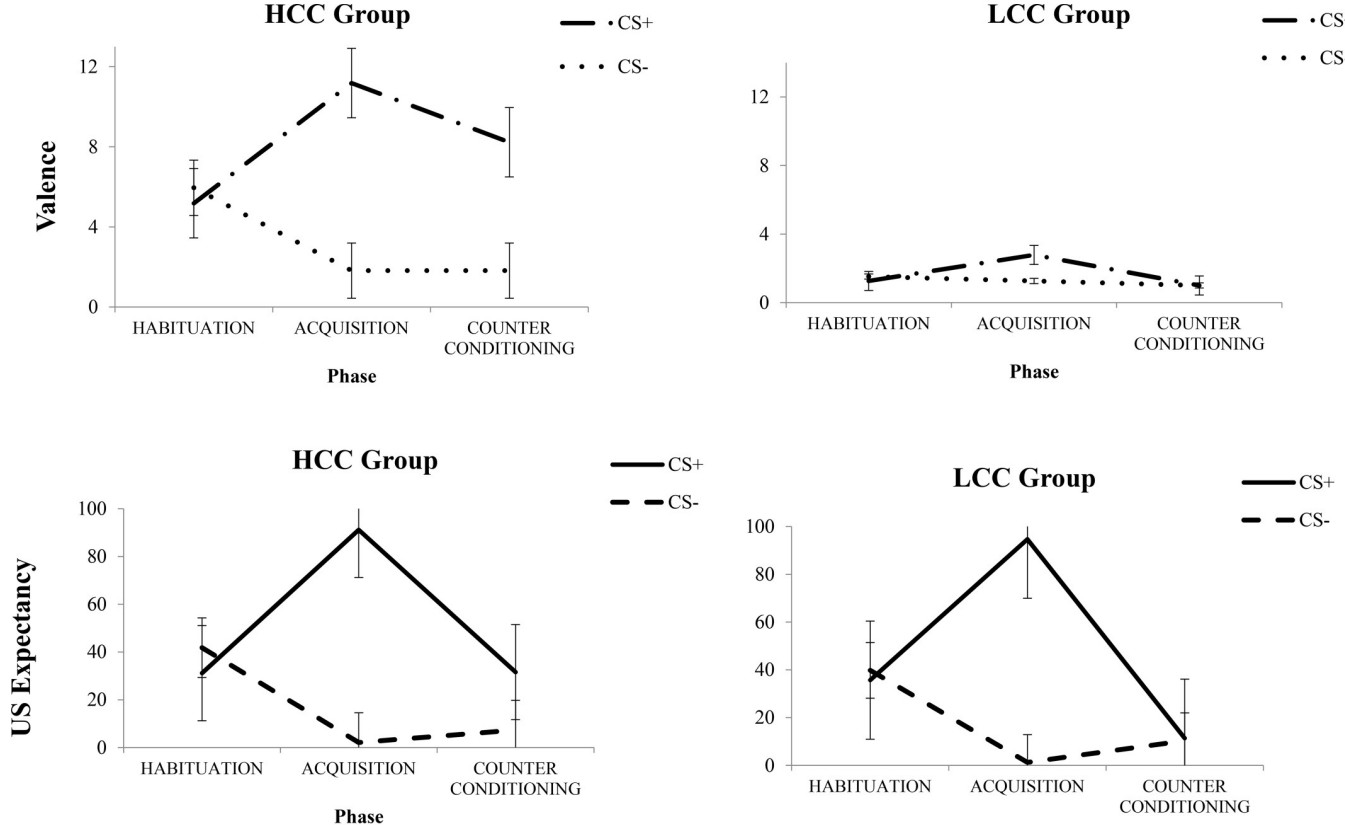

**Fig 1. Direct evaluation and US expectation ratings of the CS+.** Note: HCC: High contamination concerns group, LCC: Low contamination concerns group.

**Disgust ratings.** The main effect of CS $F(1,43) = 8.20$, $p = .006$; $n^2_p = .16$ was significant. Participants experienced more disgust in response to the CS+ compared to the CS-. The main effect of group $F(1,43) = 4.75$, $p = .03$, $n^2_p = .10$ was significant, and qualified by a significant CS by group interaction $F(1,43) = 4.25$, $p = .04$; $n^2_p = .09$, indicating that the evaluative conditioning effect took place (Table 2, Fig 1). A t-test reveals that the HCC group reported experiencing more disgust to the conditioned CS (CS +) following the acquisition stage $t(43) = 5.17$, $p = .02$, 95% CI [3.69; 34.63] (Table 2). Thus, the HCC group reported experiencing more disgust to the CS+ following the acquisition stage.

## Counter conditioning

**US expectation rating.** The results show a main effect of CS $F(1,43) = 7.36$, $p = .01$, $n^2_p = .14$. Participants rated the US expectancy occurrence significantly higher for CS + than CS-. The main effect of group $F(1,43) = 2.32$ $p = .13$, $n^2_p = .07$ was not significant (Table 2). However, the interaction effect between CS and group $F(1,43) = 6.19$, $p = .01$, $n^2_p = .12$ was significant (Fig 1). A t-test reveals that, compared to the LCC group, the HCC group reported greater anticipation of the US during CS+ presentation $t(43) = 2.49$, $p = .01$, 95% CI [3.88; 36.51]. In the analysis of the evolution of the conditioned response according to the phases of the procedure (acquisition, counter-conditioning), we observed a significant interaction effect of CS with phase $F(1,43) = 241.82$, $p < .001$, $n^2_p = .73$. Paired t-tests revealed that the US expectation associated with CS + decreased significantly from post- acquisition to post-counter-conditioning procedure $t(43) = 5.04$, $p = .005$, 95% CI [3.61; 39.90], whereas the US expectation for CS- did not change significantly $t(43) = 2$, $p = .46$, 95% CI [-14.40; 6.68] (Fig 1). The counter-conditioning procedure thus has extinguished the conditioned US expectation response in both groups of participants. The CS by phase interaction was further qualified by a CS by stage by group interaction, $F(1,43) = 7.46$, $p = .008$, $n^2_p = .08$. Compared to the LCC group, the HCC group reported a smaller reduction in the US expectation occurrence following CS + from post-acquisition to post counter-conditioning $t(43) = -2,96$ $p = .005$, 95% CI [7.61; 39.90] (Fig 1). A similar effect was not observed for the CS- $t(43) = 0,44$; $p = 0.66$ [-14.40; 6.68]. Thus, the HCC group exhibited reduced extinction of US expectancy for the CS+, specifically.

**Disgust ratings.** We observe a main effect of the CS $F(1,43) = 6.88$ $p = .01$, $n^2_p = .12$. Participants experienced more disgust in response to the CS+ compared to the CS-. The main effect of group $F(1,43) = 12.98$ $p = .001$, $n^2_p = .23$ was significant (Table 2) and qualified by an interaction effect between the CS and the group $F(1,43) = 4.82$ $p = .03$, $n^2_p = .10$. Compared to the LCC group, the HCC group reported more disgust in response to the CS+ $t(43) = 2.10$ $p = 0.04$, 95% CI [0.30; 14.4], but not the CS-, $t = 1.86$ $p = 0.07$, 95% CI [-0.06; 1.70] (Table 2, Fig 1). In the analysis of the evolution of the conditioned response according to the phases of the procedure (acquisition, counter-conditioning), we observe a CS by phase interaction, $F(1,43) = 8.20$ $p = <001$, $n^2_p = .12$. Paired samples t-tests revealed that disgust during the CS+ decreased from post-acquisition to post-counter-conditioning, $t(43) = 2.19$, $p = 0.03$, 95% CI [0.19, 4.51], whereas disgust for the CS- did not change, $t(43) = 0.44$, $p = 0.66$, 95% CI [-0.47, 0.73] (Fig 1). The CS by phase interaction was not qualified by a significant CS by phase by group interaction, $F(1,43) = 4.25$ $p = 0.05$, $n^2_p = .09$. Thus, partial extinction of disgust to the CS+ occurred: the disgust reported for the CS+ decrease significantly for the HCC group but remain significative. (Fig 1).

## Regression analyses between acquisition and reduction of US expectation and evaluative learning

Two regressions analyses were performed entering simultaneously as predictors the Disgust propensity and sensitivity scores, as well as contamination concerns (OCI-R contamination,

DOCS Contamination, and PADUA Contamination), negative affects (PANAS negative) and anxiety trait scores (STAI B) on the two following dependent variables: US expectancy ratings of the conditioned CS and the negative valence ratings of the acquisition phase. Together, the seven predictor variables explained a significant proportion of the variance in expectancy acquisition of the conditioned CS ($r^2$ = .47, F(1,40) = 11.5, p = .002) and the negative valence acquisition ($r^2$ = .45, F(1,40) = 10.62, p = .002) assessed with the EVS. The unique significant predictor for evaluative acquisition score was Disgust sensitivity ($\beta$ = 1.52, t = 3.4, p = .002). Whereas, expectation ratings after counter conditioning procedure are only predicted by anxiety scores as assessed by the STAI B ($\beta$ = .89, t = 3.36, p = .002).

## Discussion

The present study sought to test the effects of a counter-conditioning procedure on disgust extinction in two groups of participants high (HCC) and low (LCC) in contamination concerns, using a differential associative learning task that dissociated predictive learning from evaluative learning. We also tested the hypothesis that the evaluative learning of disgust that could characterize the preoccupation about dirtiness and contamination is predicted by high levels of disgust propensity and/or sensitivity.

The main findings can be summarized as follows: First, as hypothesized, the results shown that the conditioned disgust response (the valence acquired by the conditioned stimulus is correlated with self-reported level of disgust. Then, as hypothesized, the counter conditioning procedure was successful to reduce the intensity of the disgust acquired by the conditioned CS and was successful to completely extinct the US expectation. The results show that the differential conditioning procedure was successful in enhancing the disgust response toward the CS + for the HCC group but not for the LCC group. Unlike the work conducted by Armstrong in 2017, differences between groups could be seen as soon as the conditioned response was acquired. The group with strong concerns (HCC) significantly attributed to the neutral image a more negative valence as a result of its association with disgusting images. However, Schienle et al. [47] has already described in a blood phobia sample that some disgust sensitive people may be more prone to acquire disgust reaction than controls. This is also consistent with some work suggesting that attentional biases and pre-sensory learning would be an important trait in people with OCD and could foster learning and facilitate creation of disgust conditioned response, whereas people who do not have OCD may not perceive anything. This may increase their susceptibility to disgust [33]. The differences with Armstrong's work could also be partly attributed to the descriptive characteristics of our sample which is not exclusively made of students, and with equal proportion of males and females. In view of the lack of normative data regarding the disgust value and its arousal we can also hypothesize that the difference observed in the valence attracted to the conditioned CS could be partially attributable to the intensity of the disgusting stimuli used in the conditioning procedure. Moreover, the CS used by Armstrong were faces whereas we used abstract pictures tested in a previous pilot study.

Next, we found that the most robust predictor of disgust acquisition and maintenance of disgust response is not disgust propensity but disgust sensitivity, even when anxiety and negative affect are controlled. Yet it has been previously shown that disgust propensity was highly correlated with OCD, while disgust sensitivity was more generally associated with anxious disorders and inability to regulate emotion [48]. That being said, as mentioned above, many studies that have investigated this issue only evaluated DP and did not include a measure of DS in their assessments [49,50]. The impairment of meta-emotional processes could thus represent a promising perspective of investigation.

From the standpoint of US expectation, both groups have acquired and reduce their US expectations by counter conditioning procedure in the same way. The learning of the occurrence of an unpleasant stimulus is therefore not specifically acquired by people with a strong sensitivity regarding dirtiness and contamination. This is in line with Pavlovian conditioning theories: the repetition of co-occurrences produces the expectation of the unconditional stimulus following a conditioned stimulus and the repeated exposure without consequences deconstructs the expectation. Yet, the absence or the presence of occurrences can be easily noticed, which is not the case with attributed valence.

However, an important result was highlighted by the regression analyses which reveal that the US expectation and acquired valence do not rely on the same emotional processes: the US expectation is not explained by self- reported disgust but is independently explained by anxiety level, while valence acquired is exclusively explained by disgust sensitivity. Furthermore, the US expectation is totally extinguished by the counter-conditioning procedure. Thereby, the results underline the importance of the distinction between expectation learning and evaluative learning [51]: the counter-conditioning procedure reduced the evaluative conditioning effect although the levels of disgust attributed to conditioned CS remain high in the HCC group. Whereas the US expectation as is explained by the anxiety level is more completely extinguished by the procedure. Which is consistent with the fact that mere exposure succeeds to reduce it. Be as it may, beside the negative valence allocated to valanced stimuli and as positive stimuli were used in the study, ask for the pleasantness of those should have been interesting information to collect and analyze. Moreover, some works have observed that more exposure trials were necessary to reduce disgust than to reduce fear, that disgust would have a longer lasting form of learning, it would be modifiable but with difficulty [24]. We could thus suggest that more counter-conditioning trials could be effective to turn the valence off.

As hypothesized, our results show that the counter conditioning procedure is successful at reducing both the acquired valence and the predictive value of the stimulus. The two consequences of conditioning are, in this way, addressed.

Moreover, the procedure's failure to completely turn the negative valence off could be explained by the strength needed by the positive new association. Indeed, research indicates that counter conditioning requires an unconditional stimulus intense enough to be effective [52]. These preliminary results are therefore promising but the implementation of this procedure would probably require suitable desirable unconditional stimuli for each individual's situation.

From a clinical point of view, exposure-based treatment incorporating counter conditioning would thus be more effective than simple exposure treatment by addressing not only the predictive value of the unconditioned stimulus occurrence but also by modifying the disgust attributed to the conditioned stimulus (for example, wearing a beloved one's clothing during exposure, smelling or tasting something one likes, hearing a loved music. . .). Nevertheless, it will be essential to effectively observe the benefit of counter conditioning by comparing it directly with a simple exposure procedure in order to be able to draw conclusive results.

## Limits and perspectives

Although this study has addressed a certain number of limitations of previous studies such as the link between specific disgust evaluations with counter conditioning procedure, a certain number of limits can be put forward and will have to be addressed in future studies. Firstly, the analogue sample used in this task may not be fully representative of the functioning of people with obsessive-compulsive disorder in its contamination dimension. Although there is strong evidence that subclinical contamination research may be relevant to generalize results to

diagnosed patients [5], research specifically targeting clinical populations will be needed. Furthermore, it is important to note that the sample size of the included participants in the study was relatively small although sufficient to demonstrate significant effects of a counter conditioning procedure on disgust extinction in OCD patients with high contamination concerns. Therefore, our findings need to be replicated in larger samples.

Then, the present study does not directly compare the effects of the counter conditioning procedure with the effects of other therapeutic strategies that have been described in the literature as potentially effective. Indeed, although disgust has been labeled as "resistant" to extinction, some works shade this statement by describing it as modifiable, though with difficulty [53,54]. It might thus be necessary to test the differential influence of counter conditioning by requiring the inclusion of a reference condition to be sure that any difference found could indeed be attributed to differential sensitivity to counter conditioning (differential sensitivity in people with high versus low contaminations concerns). Moreover, the re-evaluation of the unconditional stimulus has been proposed as an effective technique to desensitize reactions acquired by evaluative conditioning. However, this technique has not been directly tested on disgust. Future studies will have to take into account these different techniques by directly comparing their effects with those of the counter conditioning.

In addition, self -report as common variance method may have inflated the correlations among the study variables. Future studies would benefit from using less direct and less subjective observation techniques such as emotional priming, behavioral avoidance task, eye tracking or physiological measures. Then, we did not measure pleasantness valence of the stimuli used for our evaluative conditioning procedure. Indeed, OCD has been associated with increased levels of depression [55], with in turn has been associated with a reduced reactivity to emotional stimuli [56]. Taken together the difference found in our study between HDCC and LDCC coud be explained by a reduced emotional reactivity to pleasantness between our two groups.

Moreover, a recent study using *Fmri* in fear counter-conditioning, has pointed out that implicit association could reduce the emotion generated by a conditioned stimulus [57], and could prove an interesting perspective of research.

Finally, counter conditioning, as mere extinction, implies a new learning rather than reduction or forgetfulness, and the conditioned response is always likely to return [58,59]. Because of the lack of empirical research evaluating the effects of long-term counter conditioning, we do not know if the associations acquired by evaluative conditioning also retain this original association after this type of procedure.

## Conclusion

The heterogeneity of obsessive-compulsive disorder presentations encourages us to develop new approaches that lead to more targeted management of the various processes involved. Recognition of emotional influences beyond anxiety in OCD support the critical importance of a dimensional approach of psychopathology. The data obtained here emphasizes that the inclusion of disgust and the specificity of its learning in the theoretical models of OCD, in particular C-OCD, may improve our understanding and treatment of this disorder.

## Author Contributions

**Conceptualization:** Caroline Novara, Stephane Raffard.

**Data curation:** Caroline Novara.

**Formal analysis:** Caroline Novara, Cindy Lebrun.

**Methodology:** Caroline Novara, Pierre Thérouanne, Stephane Raffard.

**Project administration:** Caroline Novara, Stephane Raffard.

**Resources:** Alexandra Macgregor, Bruno Vivet, Delphine Capdevielle.

**Software:** Pierre Thérouanne.

**Supervision:** Stephane Raffard.

**Writing – original draft:** Caroline Novara, Stephane Raffard.

**Writing – review & editing:** Caroline Novara, Cindy Lebrun, Stephane Raffard.

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
