## [Decision Letter · Decision Letter 0]

16 Dec 2020

PONE-D-20-33124

Acquisition and maintenance of disgust reactions in an OCD analogue sample: efficiency of extinction strategies through a counter conditioning procedure

PLOS ONE

Dear Dr. Novara,

Thank you for submitting your manuscript to PLOS ONE. After careful consideration, we feel that it has merit but does not fully meet PLOS ONE’s publication criteria as it currently stands. Therefore, we invite you to submit a revised version of the manuscript that addresses the points raised during the review process.

We look forward to receiving your revised manuscript.

Kind regards,

Alexandra Kavushansky, PhD

Academic Editor

PLOS ONE

Journal Requirements:

2) We note that you have indicated that data from this study are available upon request. PLOS only allows data to be available upon request if there are legal or ethical restrictions on sharing data publicly. For more information on unacceptable data access restrictions, please see http://journals.plos.org/plosone/s/data-availability#loc-unacceptable-data-access-restrictions.

Reviewers' comments:

Reviewer's Responses to Questions

**Comments to the Author**

1. Is the manuscript technically sound, and do the data support the conclusions?

Reviewer #1: Yes

Reviewer #2: Yes

Reviewer #3: Yes

2. Has the statistical analysis been performed appropriately and rigorously? 

Reviewer #1: Yes

Reviewer #2: Yes

Reviewer #3: Yes

3. Have the authors made all data underlying the findings in their manuscript fully available?

Reviewer #1: Yes

Reviewer #2: No

Reviewer #3: Yes

4. Is the manuscript presented in an intelligible fashion and written in standard English?

Reviewer #1: No

Reviewer #2: Yes

Reviewer #3: No

5. Review Comments to the Author

Reviewer #1: The authors investigated disgust valence and US expectancy during habituation, conditioning acquisition, and counter conditioning for OCD analogue sample (high dirtiness and contamination concerns group; HDCC) and healthy people (Low dirtiness and contamination concerns group; LDCC) using pink and blue neutral abstract images, eight different disgusting images and eight pleasant images. They found that counter-conditioning might reduce both US expectancy and conditioned disgust valence, and reduction was lower in the HDCC group than in the LDCC group. The following points should be criticized.

Major points

1) Though the authors used eight images representing landscapes (2), family (1), animals (2), baby (1), laughing children (2) for counter conditioning, they did not measure pleasantness valence to ask participants “How much do you find this picture pleasant?.” They should discuss possibility of difference in pleasantness valence between HDCC and LDCC and add the limitation of the study.

2) The mean disgust valence scores during habituation in the report by Armstrong et al. (Behaviour Research and Therapy, Volume 93, June 2017, Pages 78-87) were 20.62 in HCC and 10.93 in LCC, respectively. In the current study, the mean disgust valence scores during habituation were 5.18 in HDCC and 1.26 in LDCC, respectively. The authors should discuss the big difference between their current study and the report by Armstrong et al. (2017) and add the limitation of the study.

3) Armstrong et al. (2017) used 8 different disgusting images depicting feces (3), vomit (3), and rotting food (2) that were selected from the International Affective Picture System (IAPS; Lang, Bradley, & Cuthbert, 2008) and online public sources. The authors used eight different disgusting images (824x618p.) depicting spoiled food (2), waste / body secretions (3), unhygienic environments (3) that have been selected from the DIRTI database (Haberkamp et al., 2017) as US. I guess the eight images in the current study were not disgusting enough for the conditioning. The authors should discuss the disgusting images in the current study compared with the images in the reports by Armstrong et al. (2017) and add the limitation of the study.

Minor points

4) Page 2, Line 47; The authors should describe what abbreviations of HDCC and LDCC stand for.

5) Page 5, Line 125; The unconditional stimulus and conditional stimulus should be changed into unconditioned stimulus and conditioned stimulus.

6) Page 6, Line 129; The authors should differ classical conditioning from operant conditioning.

7) Page 10, Line 233; A grammatical error such as “Regression analyses was” should be corrected at revision.

Reviewer #2: It needs to be proof-read carefully. It is intelligible, but with many typos, some of which involve misnaming constructs.

See attached.

Reviewer #3: Keywords

C-OCD is not a clear acronym.

Introduction

Line 85 - I believe that there is a typo here with the word “if” - the sentence is meaningless with it.

Throughout the manuscript, authors use the word “concerns” to name the phenomena that they are observing/describing. My question here is: why not calling it symptoms? Even individuals that do not fulfill diagnostic criteria for OCD could have symptoms.

Line 130 - authors should better characterize evaluative conditioning in relation to classic conditioning. What are the differences? What are the specific characteristics of EC?

Line 136 - the sentence: “Targeted disgust-based exposure has resulted in improvement over traditional exposure based on anxiety in the treatment of blood-injection injury phobia for both symptom reduction and global health measures“ needs to be referenced.

Line 145 “only few studies”. Cite those studies.

Counter-conditioning has a hyphen - authors should standardize this term throughout the entire manuscript. The title, for example, has no hyphen.

Did the authors know the work by Koizumi and colleagues (2016) in Nature Human Behavior showing counter-conditioning in fear with fMRI? It is a possible reference that could be used in the introduction, or in the discussion section, for future studies with counter-conditioning in disgust.

Koizumi A, Amano K, Cortese A, Shibata K, Yoshida W, Seymour B, Kawato M, Lau H. Fear reduction without fear through reinforcement of neural activity that bypasses conscious exposure. Nat Hum Behav. 2016;1:0006. doi: 10.1038/s41562-016-0006. Epub 2016 Nov 21. PMID: 28989977; PMCID: PMC5628620.

Materials and Methods

Lin 166 - The sample needs to be better characterized: who were these 162 “lay people’? How they were contacted? They were accessed via the internet or in person? There is a bias in the selection, for example, individuals from the university or from the hospital?

How did you get the final sample of 25 and 23 participants in each group? It was arbitrarily?

Line 195 - there is a ref with the author/date system *Haberkamp et al., 2017).

Line 204 - 2016 should be within parenthesis.

Line 206 - ITI isn’t the acronym for inter trail interval?

Why, in the habituation phase, there is a long interval (from 12 to 18 seconds)? In fMRI studies, they are used for the BOLD signal returns to its baseline. Is there a reason for this behavioral study?

Statistical plan - first the authors mention that analyses were conducted separately, so they didn’t correct for multiples comparisons (that’s ok), but after that, they mention that ratings evolution between acquisition and CC needed a within-subject factor. This was not clear to me. The ANOVA model that included stage is not a repeated measures ANOVA?

Which statistical software or package was used to run the analysis?

Also, in lines, 213, 208, and 205 authors use the word ‘phase’ to name the section of the experiment, but in the statistical plan, they changed it to ‘stage’. This confounds the reader. Only one nomenclature should be used throughout the entire manuscript.

Line 231 - typo: two final dots

Line 241 - all p-values are written with three decimals, but the last one in the sentence is with only two (this can need to be corrected in the tables as well).

Results

Line 241 - there is an extra space before the closing parenthesis.

I would like to see boxplots for each variable in Table 1 and 2 to see their distributions. Do they have normal (or approximately normal) distribution?

Table 1 - why the HDCC group has n = 22 participants in the table? I don’t know if I missed something but in the abstract (and at the beginning of the methods) authors mention that N = 25. How (and why) those three participants were excluded? Or this is a typo?

Table 2 - there is a missing SD in the valence (CS-) of LDCC group. Also, since one of the main goals of this research is the between-group comparison, I believe that Table 2 should be re-oriented (eventually broken into two tables): groups must be at the columns, side-by-side (eventually with p-values in the third column as well), with expectation in one block and valence in the other (if authors prefer, they could slipt two tables, one for expectation and the other to valence). The experiment phases should be reoriented to the lines.

Figure 1 - the acquisition phase is not named. Also, authors must include error bars for the reader to visualize the variability. It could be CI, SD, or SE.

Why not also inserting a figure for CS-? Visualizing it and comparing it to Figure 1 would be much easier than interpreting the tables. If the authors could insert the CS- in Figure 1 would be even better (since there is not too much information, of course). I also suggest breaking Figure 1 into two figures - one just for Valence, the other only for expectancy,

Overall, the results are very long and tiring for the reader - I would recommend the authors to give more emphasis on the tables and, mainly on Figures. Explaining the results thought the figures would be much better.

Line 309 - check parenthesis and comma. Also, LDCC is writing as LCC (lots of typos throughout the manuscript!). This makes it even harder for the reader to understand the long results and the many analyses performed.

Lines 306 and 309 - difference averages don’t need to be written in the text (as in this case).

Lines 335-341 I couldn’t understand if there were significant associations. In the Methods section there is no reference to correlation. If authors performed correlations, a table showing each correlation coefficient (and p-value) could be inserted as supplementary files.

Discussion

Line 370 - It is a new paragraph? I don’t see why.

The difference between using faces vs abstract pictures should be further explored in the discussion - what are the authors’ hypotheses about it? Why changing stimuli could lead to differences?

Line 396 - the authors could explain what they meant with the sentence “This knowledge must rely on associative network.”

Lines 408-415 - is it possible that more trails (or even a second-round) of CC could be effective to turn the negative valence off?

Lines 416-419 - This last paragraph could also be extended: authors could give an example of what would it be to incorporate counter-conditioning in the clinical practice (illustrating in word what is to address the predictive value and to modify the valence).

6. PLOS authors have the option to publish the peer review history of their article (what does this mean?). If published, this will include your full peer review and any attached files.

Reviewer #1: No

Reviewer #2: No

Reviewer #3: No

---

## [Author Response · Author response to Decision Letter 0]

26 Apr 2021

We warmly thank the reviewers for their time and interest in reviewing this manuscript.

We have carefully considered the Reviewers’ and Editor’s comments and have revised the manuscript accordingly. We provide a detailed explanation of how the manuscript was modified in the files uploaded with the manuscripts and the cover letter.

---

## [Decision Letter · Decision Letter 1]

2 Jun 2021

PONE-D-20-33124R1

Acquisition and maintenance of disgust reactions in an OCD analogue sample: efficiency of extinction strategies through a counter conditioning procedure

PLOS ONE

Dear Dr. Novara,

Thank you for submitting your manuscript to PLOS ONE. After careful consideration, we feel that it has merit but does not fully meet PLOS ONE’s publication criteria as it currently stands. Therefore, we invite you to submit a revised version of the manuscript that addresses the points raised during the review process.

Please address the comments of the Reviewer #3. Please have the manuscript thoroughly processed by a professional English language editor. 

We look forward to receiving your revised manuscript.

Kind regards,

Alexandra Kavushansky, PhD

Academic Editor

PLOS ONE

Journal Requirements:

Reviewers' comments:

Reviewer's Responses to Questions

**Comments to the Author**

1. If the authors have adequately addressed your comments raised in a previous round of review and you feel that this manuscript is now acceptable for publication, you may indicate that here to bypass the “Comments to the Author” section, enter your conflict of interest statement in the “Confidential to Editor” section, and submit your "Accept" recommendation.

Reviewer #1: All comments have been addressed

Reviewer #2: All comments have been addressed

Reviewer #3: (No Response)

2. Is the manuscript technically sound, and do the data support the conclusions?

Reviewer #1: Yes

Reviewer #2: Yes

Reviewer #3: Yes

3. Has the statistical analysis been performed appropriately and rigorously? 

Reviewer #1: Yes

Reviewer #2: Yes

Reviewer #3: Yes

4. Have the authors made all data underlying the findings in their manuscript fully available?

Reviewer #1: Yes

Reviewer #2: Yes

Reviewer #3: No

5. Is the manuscript presented in an intelligible fashion and written in standard English?

Reviewer #1: Yes

Reviewer #2: Yes

Reviewer #3: No

6. Review Comments to the Author

Reviewer #1: The authors have adequately responded to the reviewer's comments and provided point-by-point answers on their article.

Reviewer #2: The authors did a thorough job addressing the concerns of the reviewers. I suggest accepting the manuscript.

Reviewer #3: Thank you for the responses - the new version of the manuscript is improved. However, there are still many typos along the text and some of my previous questions/appointments were not corrected. Authors should be more careful if they really are planning to publish these results. My comments are detailed low

Authors have to choose only one way of reporting counter conditioning: with or without the hyphen, all together as in “counterconditioning”, and apply it through the entire manuscript.

Do not start the introduction with an acronym.

“Cognitive Behavioral Therapy” does not need capital letters.

Pavlovian Conditioning – one time is written with capital letters and the second time only Pavlovian is with a capital letter (the second is right).

Pg 5, line 13 – there is an extra dot.

Pg 5, line 15 – refs should be [21-24] not [21- 23-24]

Pg 5, line 24 – ref should be [8, 21] not [8-21]

In the text, PI should be PI-C (following the acronym that first appear in the text).

Pg 7, line 21 there is an extra parenthesis, or parenthesis is lacking. In the same sentence – there is no need to start the scale’s name with capital letters.

Too many typos for a revised version of the manuscript! The authors should review the entire text.

Methods

Nice that authors incorporated the Bonferroni correction – however that rationale for using it was not described – i.e. how many comparisons are the authors controlling for?

The Bonferroni was used only for the regressions or the other analyses as well? Please be more specific and clear regarding this issue.

Results

Pg 11, line 8 – an extra parenthesis in the text.

Table 2 – the information between parenthesis is SD? Why there are no between-group comparisons in this table? This should be informative for the reader.

Also, I couldn’t understand the 1 and (0) in the last two lines of the LCC column.

Authors could review this information, Pg 14, line 16: t(43)=5.17, p=.02, 95% CI [-.47; 1.5]. If the CI is crossing zero, then it should not be significant.

What information the IC was meant to show in the results (they only appear in the t-test comparison), please clarify.

The results section is very long and the reader gets tired of reading. It is hard to follow so many analyses – Figure 1 should guide the results. Much more easy to interpret the Figure than the pages and pages of p-values and different comparisons. Authors should think of a way of passing the information that is more direct and clear and presented it in the new version of the manuscript.

Also about figure 1 - HCC is on the left in the upper part, but at the right in the bottom part. Standardize this to facilitate the reader’s life!

At the end of the paragraph about the regression analyses is was not clear to me why de disgust sensitivity was represented with Beta, t, and p-value, whereas the STAI-B has r², f, and p-value. Both analyses shouldn’t have the same parameters (as they were similar). What is the beta value of STAI-B?

Discussion

Pg 17, line 26 – there is an open parenthesis that is not closed. The third time that this happens in the manuscript.

P 18, line 10 “This is also consistent with some work suggesting that attentional biases and pre-sensory learning would be an important trait in people with OCD and could foster learning and facilitate the creation of disgust conditioned response, whereas people who do not have OCD may not perceive anything.” Authors could cite this work. Is it ref #33?

Regarding the previous response, I’m still not satisfied with the boxplots shown in the pdf – many of them were cut, the x-axis title was not readable. Most importantly, in the tables presented in their response, there are 24 participants for the LCC group and 24 for the HCC group, whereas in table 1 of the main manuscript the number of the LCC is 23 and the main text in the “participants section” the authors wrote that their HCC sample was 25. This is confusing!

7. PLOS authors have the option to publish the peer review history of their article (what does this mean?). If published, this will include your full peer review and any attached files.

Reviewer #1: No

Reviewer #2: No

Reviewer #3: No

---

## [Author Response · Author response to Decision Letter 1]

23 Jun 2021

Dear editor and reviewer

We have carefully considered the Reviewers’ comments and have revised the manuscript accordingly. We provide a detailed explanation of how the manuscript was modified in an attached file.

Sincerely yours.

---

## [Editor Report · Decision Letter 2]

30 Jun 2021

Acquisition and maintenance of disgust reactions in an OCD analogue sample: efficiency of extinction strategies through a counter-conditioning procedure

PONE-D-20-33124R2

Dear Dr. Novara,

We’re pleased to inform you that your manuscript has been judged scientifically suitable for publication and will be formally accepted for publication once it meets all outstanding technical requirements.

Kind regards,

Alexandra Kavushansky, PhD

Academic Editor

PLOS ONE
---

## [Editor Report · Acceptance letter]

2 Jul 2021

PONE-D-20-33124R2 

Acquisition and maintenance of disgust reactions in an OCD analogue sample: efficiency of extinction strategies through a counter-conditioning procedure 

Dear Dr. Novara:

I'm pleased to inform you that your manuscript has been deemed suitable for publication in PLOS ONE. Congratulations! Your manuscript is now with our production department. 

Kind regards, 

on behalf of

Dr. Alexandra Kavushansky 

Academic Editor

PLOS ONE